# Research on the Relationship between Empathy, Belief in a Just World, and Childhood Trauma in Pre-Clinical Medical Students

**DOI:** 10.3390/healthcare10101989

**Published:** 2022-10-11

**Authors:** Yanan Zheng, Die Hu, Xiaoyu Li, Mei Yin

**Affiliations:** 1School of Humanities and Social Sciences, Harbin Medical University, Harbin 150081, China; 2Medical History in Central Soviet Area and Healthy China Research Center, Gannan Medical University, Ganzhou 341000, China

**Keywords:** pre-clinical medical students, empathy, belief in a just world, childhood trauma

## Abstract

Objective: To explore the relation between empathy, belief in a just world, and childhood trauma in pre-clinical medical students. Method: Answers were collected from 880 pre-clinical medical students to questionnaires such as the Jefferson Scale of Physician Empathy-Students Version (JSPE-S), the Belief in a Just World Scale, and the Childhood Trauma Questionnaire, in November 2021. Results: The empathy of pre-clinical medical students was positively correlated with their belief in a just world (*r* = 0.194, *p* < 0.01) and negatively correlated with their childhood trauma (*r* = −0.256, *p* < 0.01); the mediating effect analysis showed that belief in a just world had a partial mediating effect on empathy and childhood trauma. Conclusion: Belief in a just world plays a mediating role between empathy and childhood trauma among pre-clinical medical students.

## 1. Introduction

Empathy refers to the ability of an individual to understand emotions from the perspective of others and apply them to communication. Empathy originates in psychology, but it has been widely applied to the fields of education and medicine [1,2,3]. Empathy is not the same as sympathy, understanding, and support. It highlights helping others with a positive attitude while shouldering their responsibilities as well as understanding and considering problems from the perspective of others [4]. In the field of medical education, empathy is an essential professional quality for medical students [5]. There is still room for improvement in the doctor–patient relationship in China, and empathy is an important factor affecting the doctor–patient relationship and doctor–patient trust [4]. For cultivating medical students’ humanistic qualities such as communication ability and teamwork, it is of great significance and value to be aware of the current situation, explore the influencing factors of empathy, and improve it in a targeted way. 

Childhood trauma is a serious issue that has not yet been fully recognized by all walks of life in China. Adverse childhood circumstances may negatively affect individuals for decades or even for life [6]. A large number of studies have shown that childhood trauma negatively impacted individual mental health, emotional communication, and interpersonal trust in adulthood [7,8,9]. In addition, empathy requires that individuals understand the emotions of others. Therefore, this study hypothesized that childhood trauma has a negative effect on empathy. 

Belief in a just world can be defined as an individual’s understanding and psychological need for justice. In other words, it refers to the degree and level to which individuals believe that the world is just and harmonious and that others and the world will treat them fairly [10]. Individuals with a low belief in a just world tend to adopt “conspiratorial” and “self-directed” attitudes to explain social phenomena, which not only affects individual mental health, such as subjective well-being and anxiety [11,12], but also affects individual communication skills such as interpersonal sensitivity [13] or the level of difficulty for individuals to be able to think or experience life ‘in other people’s shoes’ and understand others’ emotions. Based on this, this study hypothesized that belief in a just world positively impacts empathy. 

Furthermore, belief in a just world is associated with individual past experiences, especially childhood experiences. When individuals experience severe childhood trauma, it is often difficult for them to look at others and the surrounding environment from an objective and fair perspective, which affects their level of belief in a just world [14]. In other words, childhood trauma negatively impacts belief in a just world. Belief in a just world is associated with empathy and traumatic childhood experiences, and whether there is an additional effect is worth exploring further. Based on this, this study hypothesized that belief in a just world mediates the relationship between childhood trauma and empathy. 

In China, medical students usually enter the hospital for clinical practice in the last year before graduation, which is the stage at when they combine theoretical knowledge with clinical practice. In this study, medical students who are still learning theoretical knowledge in school and have not yet gotten into clinical practice are defined as pre-clinical students. The empathy level of pre-clinical medical students directly affects their professional qualities in the subsequent clinical practice stage, such as doctor–patient communication and professional identity. To further explore the relationship between pre-clinical medical students’ belief in a just world, childhood trauma, and empathy, with the aim of providing references for theoretical learning in school and the reform of humanistic medical education in subsequent clinical practice teaching, such as empathy cultivation, the research group conducted a questionnaire survey among 880 pre-clinical medical students in Gannan Medical College in November 2021, which is reported as follows.

## 2. Materials and Methods

### 2.1. Study Design, Sample, and Participants

In this study, cluster sampling was used to select participants from Gannan Medical University’s third-year and fourth-year medical students majoring in clinical medicine. The clinical medicine major at Gannan Medical College is a five-year program. As freshmen and sophomores, they are taking basic medicine courses, but have not yet begun studying humanistic medicine courses. Thus, they lack sufficient knowledge of doctor–patient communication skills, medical students’ professional qualities, and the medical environment. In the third or fourth grade, students have completed basic medicine courses and are taking courses in clinical medicine. In addition, the third-year students are studying humanistic medicine courses, such as doctor–patient communication, introduction to clinical medicine, medical sociology, etc.; the fourth-year students have completed the courses of humanistic medicine, so the third-year and fourth-year students have a certain understanding of communication ability, the medical environment, and medical students’ professional qualities. The fifth-year medical students are in clinical practice at the hospital. According to the purpose of this study, which is to investigate the relationship between empathy, childhood trauma, and belief in a just world among pre-clinical medical students, fifth-year students who are in clinical practice are excluded from the study, and since freshmen and sophomores lack a certain understanding of communication skills, the medical environment, and medical students’ professional qualities, they were excluded. Therefore, only the third and fourth graders from Gannan Medical College were recruited for this study. 

To ensure that all participants were third- and fourth-year medical students at Gannan Medical University, the research team contacted teachers of the third and fourth grades and requested assistance in distributing and retrieving questionnaires during recess as well as obtaining informed consent from participants. A total of 902 questionnaires were sent out, of which 895 were recovered, making a recovery rate of 99.22%; 880 questionnaires were valid, making a validity rate of 97.56%. All participants were between the ages of 18 and 24. Among them, 419 were boys and 561 were girls. The sample consisted of 441 third-year medical students and 439 fourth-year medical students.

The inclusion criteria were: (1) Students who consented to participate in the study with full knowledge of its purpose and content, and (2) Students who had a certain understanding of communication skills, the medical environment, and the professional qualities of medical students. The exclusion criteria were: (1) Students who were absent during the investigation period, and (2) Students who were suffering from mental illness or who were taking psychotropic medication during the investigation period. 

### 2.2. Questionnaire

#### 2.2.1. Jefferson Scale of Physician Empathy-Student Version (JSPE-S) 

Jefferson Scale of Physician Empathy-Student Version was used to evaluate the empathy level of participants. The scale was originally compiled by Hojat, et al., and Wan Xiaoyan, Liu Yuanyuan, and Jiang Tian, et al. modified this scale to make it suitable for evaluating empathy in medical and nursing groups in China based on the Chinese culture and language [15]. The scale consists of 20 items on a seven-point scale from ‘strongly disagree’ to ‘strongly agree’. The overall score is the sum of the answers. The higher the score, the higher the respondents’ empathy level. 

#### 2.2.2. Belief in a Just World Scale

The Belief in a Just World Scale was used to evaluate the degree of participants’ belief in a just world. Based on Dalbert’s original scale, Zhang Dajun and Su Zhiqiang, et al. modified the scale, according to the Chinese cultures and languages, in order to better assess the level and degree of belief in a just world among various groups of people in China [16]. The scale consists of 13 Likert-type questions on a point scale of 1 (strongly disagree) to 6 (strongly agree). A total score is calculated by adding up all the answers. As the score goes up, the stronger is the respondents’ belief in a just world, thus, the more they believe that they will be treated fairly and justly by the surrounding group and the more inclined they are to believe that their efforts will be rewarded accordingly. 

#### 2.2.3. Childhood Trauma Questionnaire 

The Childhood Trauma Questionnaire was used to evaluate the degree of childhood trauma of participants. This questionnaire was developed originally by Bernstein, et al. and modified later by Zhang Yalin and Zhao Xingfu, et al. based on Chinese culture and language, in order to be more suitable for assessing children’s childhood trauma in China. At present, the questionnaire is widely used among Chinese college students [17]. It includes five dimensions that can provide a comprehensive assessment of respondents’ experiences of childhood trauma, including physical neglect, physical abuse, emotional neglect, emotional abuse, and sexual abuse. The questionnaire includes 28 items using a 5-point Likert scale ranging from 1 (never) to 5 (always). The overall score equals the sum of item scores. The higher the total score, the more serious the childhood trauma experience of the respondents. This study evaluated the level of childhood trauma experience of participants with a total score of the questionnaire.

### 2.3. Data Analysis

Data management and analysis were performed using SPSS22.0. The relationship between childhood trauma, belief in a just world, and empathy was analyzed using the Pearson correlation analysis. The SPSS PROCESS plug-in compiled by Hayes was used to further explore the mediating effect of the belief in a just world on experiences of childhood trauma and empathy. In this study, *p* value < 0.05 was considered as a statistically significant difference.

## 3. Results

### 3.1. Common Method Variance Test

In this study, the Harman single-factor test was used for the common method variance test, and the results showed that the variation of the first factor obtained without rotation was 22.78%, less than the critical value of 40%. It showed that common method variance had no significant effect on this study.

### 3.2. Reliability Analysis of the Jefferson Scale of Physician Empathy-Student Version, the Belief in a Just World Scale, and the Childhood Trauma Questionnaire

In this study, the Cronbach’s α of JSPE-S was 0.839, the Cronbach’s α of the Belief in a Just World Scale was 0.901, and the Cronbach’s α of the Childhood Trauma Questionnaire was 0.822.

### 3.3. Correlation Analysis of Empathy, Belief in a Just World, and Childhood Trauma 

Empathy showed a significant positive correlation with belief in a just world, while empathy showed a significant negative correlation with childhood trauma among 880 students. In addition, a significant negative correlation was found between childhood trauma and belief in a just world. The results of the correlational analysis are set out in Table 1.

### 3.4. The Mediating Role of Belief in a Just World in Empathy and Childhood Trauma among Students

The correlation analysis revealed that there was a statistically significant correlation between empathy, belief in a just world, and childhood trauma. SPSS Process plug-in was used to examine whether there was a mediating effect between the three variables. In this study, the empathy of the respondents was taken as the dependent variable, childhood trauma as the independent variable, and belief in a just world as the intermediary variable. According to the Bootstrap method of Hayes [18], “A sample size of 5000 was chosen with a confidence interval of 95%”, was used to test whether there was a mediating effect between the three variables. The results showed the mediating interval between belief in a just world and childhood trauma (LLCI = −0.064, ULCI = −0.006) did not contain 0, suggesting that belief in a just world had a significant mediating effect between empathy and childhood trauma. In addition, after controlling the mediating variable (belief in a just world), the independent variable (childhood trauma) had a significant effect on the dependent variable (empathy), and the interval (ILCI = −0.672, ULCI = −0.498) did not contain 0, suggesting that belief in a just world played a partially mediating role in the relationship between empathy and childhood trauma of participants. The mediating effect ratio was 5.25%.

## 4. Discussion

Empathy is an important part of medical students’ professional quality and humanistic spirit. Knowing the factors that influence students’ empathy and improving them in theoretical study and subsequent clinical practice teaching is of great significance and value in order to cultivate excellent medical staff, heighten communication ability, and improve the doctor–patient relationship. The result of the survey shows that students who believe that the world is just and fair and those who have fewer traumatic experiences in childhood have higher levels of empathy. This finding is similar to the conclusions that “medical students’ belief in a just world was positively correlated with self-compassion” [19], “alexithymia of medical students with childhood trauma experiences was more severe” [20], and so on. According to psychology, individuals who experience emotional neglect, abuse, and other traumatic experiences in childhood tend to have defective emotional expression in adulthood [21]. They find it difficult to understand others and to think from others’ perspectives, which affects their empathy level. Students with a higher level of belief in a just world tend to believe that they will be treated fairly and justly. When communicating with others, they believe that their efforts and understanding will be reciprocated accordingly, and they are therefore more likely to put themselves in another’s shoes, so their empathy level is high. In contrast, students with a low level of belief in a just world are more likely to see the world through the perspective of “prejudice” and “self-orientation” and less likely to put themselves in another’s shoes, which affects their empathy. These findings indicate that professional literacy education, responsibility awareness education, and other means, in addition to communication training, to improve the understanding and belief of justice, for students who suffered childhood trauma, can also help them better understand the feelings of others, and effectively increase their empathy level.

A more significant finding was that childhood trauma was not only negatively associated with empathy, but also that it mediated the level of empathy via belief in a just world. Generally speaking, the belief in a just world is one’s spiritual pursuit of fairness and justice, as well as an individual’s way of explaining the surrounding environment. Due to the lack of emotional experience in childhood, students with relatively severe childhood trauma are more likely to distrust others in communication. They tend to perceive other people and their surroundings as biased and unfair towards them. This tendency reduces their belief in a just world and exacerbates their hostility and distrust towards other individuals, making it difficult for them to put themselves in other people’s shoes and understand their emotional experience. Students with childhood trauma, on the one hand, tend to distrust other people due to past experiences and may hold the view that the environment is unfair to them. On the other hand, if students perceive that their surroundings make it difficult for them to receive fair treatment, it often interferes with their ability to consider problems from others’ perspectives. Overall, improving the empathy level of medical students is of great significance for improving the doctor–patient relationship in China, and changing childhood trauma experiences is difficult. Therefore, students with childhood trauma may benefit from many strategies such as quality development training, vocational quality education, and group counseling, that can improve not only their communication skills, but also their belief in a just world, and their empathy level.

## 5. Conclusions

Based on this study, childhood traumatic experiences and belief in a just world are strongly associated with empathy among pre-clinical medical students. Education and improvement of belief in a just world may directly affect the improvement of students’ empathy level, particularly in medical education. Additionally, carrying out the training required to develop communication skills, interpersonal skills, and trust ability is necessary for students with childhood trauma experiences. Furthermore, the mechanism of belief in a just world on empathy should be explored more deeply in the future, such as whether it is influenced by other psychosocial factors, which will be of great use in improving the cultivation of humanistic quality in the medical field and professional spiritual education. Lastly, it’s worth noting that the results of this study are applicable only to the study population (Chinese pre-clinical students) and are not generalizable to other populations.

## Figures and Tables

**Table 1 healthcare-10-01989-t001:** Correlation analysis of empathy, belief in a just world, and childhood trauma.

Programs	M ± SD	Empathy	Belief in a Just World	Childhood Trauma
empathy	106.68 ± 14.02			
belief in a just world	56.05 ± 9.91	0.194 **		
childhood trauma	46.48 ± 9.86	−0.438 **	−0.256 **	

Note: ** *p* value < 0.01.

## Data Availability

The data presented in this study are available on request from the corresponding author. The data are not publicly available due to privacy reasons.

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
