# Peer review of "Research on the Relationship between Empathy, Belief in a Just World, and Childhood Trauma in Pre-Clinical Medical Students"

_healthcare, 2022, doi:10.3390/healthcare10101989_

Round 1
Reviewer 1 Report
It is interesting to explore the relationship between empathy and own beliefs. But please could you mention what your hypothesis is. You conclude that there are specific issues in China. This may need to be described early on and the objective to be related to these issues. The objective is not well described and it is probably incorrect to claim that relationship is being explored as there is no evaluation of bidirectionality or adjustments for confounding factors.
Please describe the population better. You mention early on 'non-practice medical students', which is not clearly explained until much later, in the 'Methods'. There is a repetition of the term several times in the abstract, and multiple times throughout the text. I suggest introduce and describe your population (students not in hospital practice) and thereafter refer to as 'participants' or 'students' since you already specified that they are not in practice. Educational systems may differ and some students may be having practice in hospitals parallel with their studies, others may not be allowed to practice before graduating medical school. I suggest also describing the population so it can be understood internationally (e.g. grade 19 and grade 18 are not readily comprehensible to all).
What do you mean by 'the scale was localized by Wan Xiaoyan..'? Do you mean that the scale you used in your study was modified/validated to be used by Chinese medical care groups by Wan X, et al? The same applies to the other questionnaires used in the study which you present in the same way. Or do you mean 'locally adapted' by... 'Localized' is unclear.
Please describe participation more clearly. You sent out 902 questionnaires (link, post, other?) How many responded and how many of these were valid? Response rate may not be the same as participation rate, for instance, if all responded positively but many did not answer the questionnaires properly vs all who responded also participated with valid answers to all questionnaires.
There are several typos in the abstract. Please read through. The absract should be meticulous. Examples of typos in the Abstract: 'Date' from 880... Change to 'Data' from 880...; ...collected via... change to for instance 'collected answers to questionnaires, such as....; spacing where it does not belong.
Reviewer 2 Report
Title: do you mean non-practicing instead of non-practice? Or perhaps, pre-clinical? Or perhaps medical school students?
Line 55-57: the sample was recruited from 9 classes. What percentage of the classes participated? And were there incentives for them to participate? Were there exclusion criteria for any of the members of the classes you recruited from? How do you ensure that the individuals participating cannot be personally identified?
Line 58: what is the age of the students? Are they still considered to be girls and boys or might they be considered to be women and men?
Lines 72, 73, 81, 82, 92, etc.... should contain reference numbers, rather than the names of the authors of the work being referenced.
lines 76-77, 87-88, and 100-101 are results and do not belong in the methods section.
Table 1 appears to show only the significant results. All results should be reported.
Line 128-129: I do not know what a deep relation is. Could you define?
Line 132: how did you get a sample size of 5,000?
Round 2
Reviewer 1 Report
The authors have adequately addressed the comments.
I have only a minor comment: Please point out that your results are applicable only for your study populatieon (Chinese pre-clinical students), and are not generalizable for other populations. Your hypothesis is now clear, but all results will still only be true for the studied population and can not automatically be generalized to other populations.
Author Response
Thanks for your suggestion. We added the section you mentioned and put it in the conclusion section. The corresponding changes are as follows. Major modifications are highlighted in red font.
- Conclusion
Based on this study, childhood traumatic experiences and belief in a just world are strongly associated with empathy among pre-clinical medical students. Education and improvement of belief in a just world may directly affect the improvement of students' empathy level, particularly in medical education. Additionally, carrying out the training required to develop communication skills, interpersonal skills, and trust ability is necessary for students with childhood trauma experiences. What’s more, the mechanism of belief in a just world on empathy should be explored more deeply in the future, such as whether it is influenced by other psychosocial factors, which will be of great use in improving the cultivation of humanistic quality in the medical field and professional spirit education. Lastly, it’s worth noting that the results of this study are applicable only to the study population (Chinese pre-clinical students), and are not generalizable to other populations.
Reviewer 2 Report
Thank you, authors, for responding promptly and completely to suggestions for change.
Author Response
Thank you for your valuable advice before, which was of great help to us.